# Research Progress on Using Nanoparticles to Enhance the Efficacy of Drug Therapy for Chronic Mountain Sickness

**DOI:** 10.3390/pharmaceutics16111375

**Published:** 2024-10-26

**Authors:** Boshen Liang, Yang Zhou, Yuliang Qin, Xinyao Li, Sitong Zhou, Kai Yuan, Rong Zhao, Xiaoman Lv, Dongdong Qin

**Affiliations:** 1Key Laboratory of Traditional Chinese Medicine for Prevention and Treatment of Neuropsychiatric Diseases, Yunnan University of Chinese Medicine, Kunming 650500, China; 15546459039@163.com (B.L.); zhou_yang113@163.com (Y.Z.); q987302473@163.com (Y.Q.); lxy9081720@163.com (X.L.); 18725261705@163.com (S.Z.); 2First Clinical Medical College, Yunnan University of Chinese Medicine, Kunming 650500, China; zhaorongkm@163.com; 3Second Clinical Medical College, Yunnan University of Chinese Medicine, Kunming 650500, China; 13051552385@163.com

**Keywords:** chronic mountain sickness, nanoparticles, drug delivery, targeted therapy, efficacy

## Abstract

Chronic mountain sickness (CMS) poses a significant health risk to individuals who rapidly ascend to high altitudes, potentially endangering their lives. Nanoparticles (NPs) offer an effective means of transporting and delivering drugs, protecting nucleic acids from nuclease degradation, and mediating the expression of target genes in specific cells. These NPs are almost non-toxic and easy to prepare and store, possess a large surface area, exhibit good biocompatibility and degradability, and maintain good stability. They can be utilized in the treatment of CMS to enhance the therapeutic efficacy of drugs. This paper provides an overview of the impact of NPs on CMS, discussing their roles as nanocarriers and their potential in CMS treatment. It aims to present novel therapeutic strategies for the clinical management of CMS and summarizes the relevant pathways through which NPs contribute to plateau disease treatment, providing a theoretical foundation for future clinical research.

## 1. Introduction

Chronic mountain sickness (CMS), also known as Monge’s disease, is a prevalent and progressive syndrome affecting many high-altitude regions around the world. Over 140 million people live above 2500 m globally, with an estimated 5–10% at risk of developing CMS [1]. This disorder, first described by Carlos Monge Medrano, primarily affects populations in the South American Andes and the North American Rockies [2]. According to the 2018 Lake Louise Acute Mountain Sickness Score, the diagnostic criteria for acute mountain sickness (AMS) include headache, weakness, gastrointestinal symptoms, and dizziness/light-headedness [3]. Currently, the most effective treatment for CMS is relocation to lower elevations. Medication may be prescribed for those unable or unwilling to migrate [4]. However, many of these drugs have significant side effects. For example, acetazolamide may cause sensory abnormalities in the limbs, frequent urination, difficulty swallowing, and taste disorders, while dexamethasone can lead to gastrointestinal irritation, hyperglycemia, and psychiatric disorders [5]. Consequently, the search for safe, effective, and convenient alternative therapies remains a key focus of high-altitude medicine research due to the limitations of Western medicines regarding accessibility and safety.

Nanoparticles (NPs) are versatile drug carriers developed to transport various therapeutic agents into cells. The advancement of NPs offers powerful tools and new directions for medical diagnostics. Since the 21st century, nanotechnology has become a prominent area of research, with significant progress in the development and application of NPs and nanodevices [6]. Inorganic NPs, as a novel class of drug carriers, show promising potential in achieving targeted drug delivery, controlling drug release, and improving the solubility and bioavailability of poorly soluble drugs, thereby enhancing drug efficacy. For example, non-metallic materials include mesoporous silica and carbon NPs, and metallic materials consist of silver, gold, iron oxide, and nickel oxide NPs [7]. Organic NPs include liposomes, extracellular vesicles, polylactic acid–hydroxyacetic acid (PLGA), and micelles. These NPs possess significant anti-inflammatory and antioxidant properties and have demonstrated therapeutic effects on several chronic pathological conditions [8]. Furthermore, NPs can deliver drugs directly to targeted tissues and cells, offering the potential for optimizing CMS treatment. This paper reviews recent advancements in nano-delivery strategies, focusing on inorganic NPs and organic NPs, aiming to propose new therapeutic approaches for CMS in clinical settings. Additionally, future discussions will explore research on gene and viral nanocarriers.

## 2. Pathogenesis of CMS

The hypoxia levels in high-altitude areas can cause CMS, a clinical syndrome characterized by erythropoietic polycythemia (HAPC), hypoxia, and pulmonary hypertension [9]. This condition primarily affects individuals residing more than 2500 m above sea level. Clinical manifestations of CMS include symptoms such as dizziness, headache, nausea, vomiting, and dyspnea. In severe cases, it may progress to high-altitude pulmonary edema (HAPE), high-altitude cerebral edema (HACE), and other complex conditions [10,11,12]. Long-term exposure to hypoxia can lead to pulmonary vasoconstriction and remodeling, increasing pulmonary circulatory resistance and pulmonary arterial pressure. This sequence of events often results in hypoxic pulmonary hypertension, which exacerbates the afterload of the right ventricle, causing largely irreversible pathological changes that significantly diminish the quality of life and life expectancy of high-altitude inhabitants [13]. The hypoxic environment also contributes to myocardial hypoxia, oxidative stress, inflammatory responses, fibrosis, apoptosis, and other injuries, affecting cardiac structure and function [14].

### 2.1. Hypoxia-Induced Factor Accumulation and Regulation

The development of hypoxic pulmonary arterial hypertension (HPAH) reduces the oxygen supply to pulmonary artery smooth muscle cells, decreasing intracellular mitochondrial hydrogen peroxide production and altering cellular redox signaling pathways. This change pathologically stimulates transcription factors. Hypoxia-inducible factor (HIF), composed of heterodimers of HIF-1α, HIF-2α, HIF-3α, and HIF-β subunits, plays a critical role in cellular homeostasis as a major transcriptional regulator in hypoxic environments and cellular inflammatory responses. Under aerobic conditions, the HIF-α subunit is quickly inactivated by oxygen-dependent hydroxylation by HIF proline hydroxylase (HPH), leading to its ubiquitination and subsequent proteasomal degradation. In hypoxic conditions, HPH activity is inhibited, allowing the stable accumulation of the HIF-α subunit and its dimerization with HIF-β, which triggers the transcription of genes that restore and maintain cellular oxygen availability. Animal studies have shown that mice deficient in HIF-1α exhibit reduced pulmonary vascular remodeling and pulmonary hypertension under chronic hypoxic conditions but may experience worsened right ventricular hypertrophy and remodeling [15]. Additionally, research by Hu et al. found that administering a HIF-2α inhibitor to hypoxic mice reduced mean pulmonary artery pressure and attenuated right ventricular remodeling, highlighting the significant role of HIF-2α in pulmonary vascular and ventricular remodeling responses to chronic hypoxia [15].

### 2.2. Oxidative Stress Leads to CMS

Oxidative stress represents an imbalance between oxygenation and antioxidant effects in the body, leading to the excessive production of reactive oxygen species (ROS) at levels that surpass the body’s capacity to eliminate them. ROS can easily react with various cellular structural components, affecting cellular differentiation, cellular apoptosis, and the immune system, thereby causing oxidative stress damage [16]. Sarada S et al. observed significantly increased levels of ROS, lipid peroxidation, and superoxide dismutase (SOD) in rats exposed to a low-pressure hypoxic environment at an altitude of 7620 m [17]. Studies have shown that exposure to high altitudes increases ROS production in capillary blood vessels while decreasing their antioxidant capacity [18]. These pathophysiological changes are closely associated with various CMS conditions. Nuclear factor E2-related factor 2 (NRF2) is a transcription factor that regulates the expression of several antioxidant proteins. Under physiological conditions, NRF2 and Kelch-like ECH-associated protein 1 (Keap1) bind together in the cytoplasmic lysate. Upon the occurrence of oxidative stress, NRF2 translocates to the nucleus and induces the expression of ROS-detoxifying enzymes, effectively preventing ROS accumulation. This upregulation of NRF2 in high-altitude environments has been confirmed as an effective alternative treatment for CMS [19].

### 2.3. Inflammation Leads to CMS

Rapid activation of inflammatory processes occurs during exposure to high-altitude, low-pressure, and low-oxygen environments [20]. Hartmann et al. examined inflammatory factors in the sera of volunteers exposed to sea-level and high altitudes (4350 m above sea level) at various time points. They found that only the level of interleukin-6 (IL-6) was consistently elevated, while other factors, such as interleukin-1β (IL-1β), tumor necrosis factor-alpha (TNF-α), and C-reactive protein (CRP), showed no significant changes. This suggests that IL-6 is more sensitive to high-altitude exposure [21]. Additionally, the serum levels of interleukin-2 (IL-2), interleukin-3 (IL-3), and macrophage chemotactic protein-1 (MCP-1), as well as IL-1β and TNF-α, have all been implicated in the pathogenesis of CMS and altitude adaptation. The altered inflammatory factors may serve as biomarkers for CMS [22]. TNF-α release and nuclear factor kappa-B (NF-κB) activation increased oxidative stress (OS) and inflammation, and inflamed tissues may also be hypoxic [23,24]. Furthermore, hypoxia promotes oxidative stress, a condition closely linked to the inflammatory process [25]. Possible functions and mechanisms in the pathogenesis of CMS are shown in Figure 1.

## 3. Properties of Inorganic NPs

NPs, due to their extremely small size, exhibit numerous exceptional properties that conventional materials lack, including unique optical, electromagnetic, thermodynamic, and quantum mechanical characteristics [26]. These attributes make them highly promising for a wide range of applications, particularly in the biomedical field. Nanotechnology facilitates the transformation of drugs into nanoscale particles, enhancing interactions between these drugs and cells (Figure 2 and Table 1). Furthermore, these NPs can be engineered for targeted delivery to specific lesion areas and controlled release, thereby increasing drug bioavailability, reducing required dosages and treatment duration, and minimizing adverse effects [27]. Inorganic NPs demonstrate high biocompatibility and the ability to traverse biological barriers, such as the blood–brain barrier (BBB) and the intestinal barrier, facilitating the transport of both hydrophilic and hydrophobic drugs while protecting against degradation [28]. Studies have shown that NPs possess the largest specific surface area among all known materials, enabling them to effectively transport and deliver drugs. They can also be used for radioactivity labeling or modification, making them valuable for targeted drug delivery, cell imaging, molecular detection, and more. Additionally, NPs exhibit good biocompatibility and degradability, allowing them to accumulate in specific organs with fewer side effects. Their slow-release properties can lower drug concentrations, thereby reducing toxic side effects. Furthermore, their stability makes them suitable for applications in bioimaging, antioxidants, and enhanced sustainability. These characteristics form the foundation for the use of NPs in the medical field [29]. Consequently, the unique properties of inorganic NPs suggest their potential application in the treatment of CMS.

The shape and size of inorganic nanoparticles have an important influence on the properties of nanoparticles, and the precise controls of shape and size are therefore crucial in the preparation of nanoparticles. The preparation of nanoparticles using block polymers as templates is one of the most stable methods for the precise control of nanoparticle size and shape. Generally, nanoparticles are prepared by hydrothermal methods, organic solution-phase synthesis, sol–gel, dendritic macromolecule template methods and decomposition. On this basis, Li et al. proposed a star-shaped single-molecule block copolymer used as a template for the preparation of inorganic nanoparticles, which overcomes the problem of poor stability in the traditional preparation method, has a more obvious advantage in the synthesis of hollow and core/shell structured nanoparticles, and enables more accurate regulation of nanoparticle size [30].

**Table 1 pharmaceutics-16-01375-t001:** Advantages and challenges of commonly used inorganic NPs.

No.	Nanoparticles	Properties	Biological Application	Challenges	References
1	Carbon	High strength, high biocompatibility, and good electrical and thermal conductivity	Alleviation of insulin resistance, intervention for cardiovascular disease, and antioxidant effects	Biological toxicity and release of carbon monoxide	[31]
2	Silicon Dioxide	Highly controllable treatment platform size and shape, low toxicity, and good biocompatibility	Antimicrobial, magnetic resonance contrast medium, intervention for cardiovascular disease, and separation of natural products	Easily internalized by cells, causing various diseases, such as neurodegenerative diseases	[32]
3	Gold	Optical properties, plasmon resonance properties, fluorescence properties, and adsorption properties	Novel optical probes, molecular recognition and labeling, and DNA sensors	Lower circulation and tissue clearance	[33]
4	Magnetic	Local magnetic, thermal, and mechanical effects and intrinsic catalytic activity	Targeted drug transport as well as controlled release, targeted gene therapy, magnetic induction tumor thermotherapy, and magnetic resonance imaging	Biological toxicity	[34]

## 4. Inorganic NPs in CMS

As a novel class of drug carriers, inorganic nanocarriers show considerable promise in enhancing targeted drug delivery, facilitating slow-release mechanisms, and improving the solubility and bioavailability of poorly soluble drugs, thereby increasing therapeutic efficacy. Inorganic NPs primarily consist of metals (such as gold, silver, platinum, and nickel), metal oxide NPs (including titanium dioxide, manganese dioxide, magnetite, and alumina), and semiconductors (such as silicon and ceramics) [35].

### 4.1. Carbon NPs in CMS

Carbon NPs are a class of materials characterized by unique structural and physicochemical properties, allowing for widespread applications in biosensors and tissue engineering. They exhibit excellent tensile strength (exceeding 100 GPa), high mechanical integrity, and a modulus of elasticity exceeding 1 TPa, along with electrical and thermal conductivity. Carbon NPs have been extensively used in the treatment of hypoxia-induced hypoxic pulmonary arterial hypertension (HPAH) and cardiac injury. Generally, based on their structure and size, carbon NPs can be categorized into zero-dimensional (such as fullerene, nanodiamonds, and carbon dots), one-dimensional (including carbon nanotubes, diamond nanorods, and carbon nanofibers), two-dimensional (like graphene and its derivatives, as well as diamond nanosheets), and three-dimensional NPs (which comprise composite NPs) [36].

#### 4.1.1. Carbon NPs in Pulmonary Arterial Hypertension

Hypoxic pulmonary arterial hypertension (HPAH) affects approximately 1% of the global population and is particularly prevalent among individuals over the age of 65, impacting up to 10% of this demographic. HPAH is classified as a pulmonary vascular disease characterized by progressive increases in pulmonary vascular resistance (PVR) and pulmonary artery pressure (PAP) resulting from pulmonary vascular contraction and remodeling.

Currently, carbon NPs are widely used in the treatment of pulmonary arterial hypertension (PAH) [37,38,39,40]. You’s study reported a delivery nanosystem utilizing programmed DNA self-assembling mammalian target of rapamycin (mTOR) siRNA-loaded DNA nanotubes (DNA-nts), which can be efficiently transfected into pulmonary artery smooth muscle cells (PASMCs) via endocytosis under both normal and hypoxic conditions. PASMCs are crucial in the pathogenesis of HPAH [41]. The mTOR siRNA significantly induced autophagy and inhibited cell growth, suggesting its potential therapeutic value for diseases associated with abnormal autophagy in PASMCs [42]. This study presents a safe, efficient, and cost-effective delivery platform that is easily scalable. Moreover, the assembled mTOR siRNA-loaded DNA nanotubes demonstrated relative stability, biocompatibility, and low cytotoxicity while significantly inhibiting the hypoxia-induced proliferation of PASMCs. This type of nanocarrier may serve as a promising delivery vector for managing HPAH and other chronic cardiopulmonary diseases [42].

The mechanistic target of mTOR is a critical signaling pathway in the body, encompassing various biological processes, including gene transcription, protein translation, ribosome synthesis, and apoptosis. mTOR significantly regulates protein translation, cell growth, and proliferation by integrating signals from growth factors, nutrients, and energy levels [43]. In pulmonary vascular cells, autophagy exhibits a dual regulatory mechanism in the occurrence and development of HPAH. Increased expression levels of LC3B and Beclin1 were observed in pulmonary artery endothelial cells (PAECs). Experiments showed that active LC3B in vascular cells may partially inhibit proliferation associated with vascular remodeling. This finding confirms that the effect of autophagy on the regulation of HPAH is bidirectional [44,45]. According to these results, autophagy may serve as a point of interaction for binding inorganic NPs in HPAH.

As a metallic material, it can cause damage when introduced into an organism. Morimoto’s study confirmed that exposure to fullerenes does not induce inflammation or causes only transient inflammation in the lungs. However, exposure to high concentrations of multi-walled carbon nanotubes (MWCNTs) and single-walled carbon nanotubes (SWCNTs) is associated with inflammation and granuloma formation in lung tissue [46]. The application of carbon NPs in hypoxia environments can have similarly detrimental effects. An in vitro HPAH model study demonstrated that carbon NPs under hypoxic conditions lead to the production of reactive oxygen species (ROS), increased cellular production of nitrite and nitrate, and the secretion of the pro-inflammatory mediator IL-6 [47].

#### 4.1.2. Carbon NPs in Cardiac Injury

Sustained or periodic hypoxia in an organism can lead to irreversible physiological disturbances. Hypoxia can precipitate various metabolic diseases, myocardial hypertrophy, and myocardial ischemia. Consequently, monitoring and treating organisms in hypoxic environments is crucial for human health and development. Chronic exposure to hyperbaric hypoxia in high-altitude environments can cause structural and functional changes in the right ventricle of rats [48]. Due to the interdependence of the left and right ventricles, changes in pressure and volume in one ventricle typically affect the other, ultimately resulting in total cardiac dysfunction [49].

Fullerene was discovered in 1985 and was the third allotrope of carbon after the discovery of graphite and diamond. Because of its three-dimensional spherical spatial structure, it can cause a lot of physical and chemical reactions. The study of fullerenes has been a general concern of the majority of scientific research workers, as these molecules have good antioxidant properties and are very effective for free radical scavenging [50]. Fullerene-C_60_-based particles have been shown to treat hypoxia-induced mitochondrial dysfunction in the mammalian myocardium, and pharmacokinetics and pharmacodynamics indicate that they are suitable for administration in single or multiple injections for the prevention and treatment of clinical conditions involving acute and chronic myocardial hypoxia without acute toxicity [51].

Lipid core NPs (LDE) used as carriers for the antitumor agent methotrexate (MTX) significantly increase the cellular uptake of MTX. The literature suggests that combined LDE-MTX treatment reduces vascular inflammation and the myocardial infarction (MI) area in hypoxic environments while inhibiting the development of cardiac hypertrophy and decreasing apoptosis, macrophage infiltration, and ROS production [31]. The combination of MTX and NPs is often used for treating cardiac insufficiency. Natalia’s study used paper NPs encapsulating MTX to improve cardiac insufficiency and hypertrophy in rats [52]. In addition to conventional carbon NPs, composite fibers prepared using carbon nanofibers and conductive polymers can theoretically combine the properties of both materials, improving their electrical properties. The functionality of NPs can be effectively integrated with the electrical conductivity of polyaniline (PANI) polymers, potentially endowing the materials with unique properties [53,54]. Composite NPs are also applicable in hypoxic environments. Moradikhah’s study used a carbon nanofiber polycaprolactone mat polymerized in situ with PANI as a cell-free antioxidant cardiac patch (CP). This reduced the oxidative stress generated during ischemia–reperfusion (I/R) following MI in hypoxic environments, decreased intracellular ROS content and caspase-3 mRNA levels, and mitigated the hypertrophic effects of hydrogen peroxide on H9c2 cells [55].

Prolonged hypoxia tends to cause a progressive increase in pulmonary vascular resistance and an increase in right heart afterload, leading to pathological dilatation of the right ventricle, reduced per-pulse output, decreased cardiac contractility, and impaired right ventricle–pulmonary artery coupling. Ultimately, this can result in right heart failure and death [16,56]. Cyclodextrin (CD) is a series of cyclic oligosaccharides composed of 6–12 D-glucopyranose units generated from straight-chain starch through the action of cyclodextrin glucosyltransferase. CD exhibits a unique property of having a hydrophilic exterior and a hydrophobic interior. Its hydrophobic cavity has a high capacity for molecular recognition and enrichment, enabling the selective identification of various organic and inorganic molecules. The hydrophilic exterior, characterized by good water solubility, allows CD to function as a surface functionalization agent for NPs, enhancing the dispersion of many non-water-soluble materials [57,58]. Currently, various forms of CD/CNT composites are widely used [59,60]. CD preparations as oxygen nanocarriers have been shown to improve the survival of cardiomyoblasts in hypoxic environments (by 12%-20%), thus reducing the damage from heart failure and demonstrating the potential of carbon NPs for cardiovascular therapy [61]. However, a search of the relevant literature did not reveal the use of graphene to intervene in CMS.

### 4.2. Silicon Dioxide NPs in CMS

In recent years, research and application of NPs in the biomedical field have rapidly advanced, enhancing the sensitivity and targeting of drug therapies. Therapeutic platforms based on silica nanoparticles offer several advantages, including highly controllable size and shape, low toxicity, good biocompatibility, stability under physiological conditions, and biosafety, making them suitable for clinical applications [62]. Silica NPs can be classified into nonporous and mesoporous silica nanoparticles (MSNs). MSNs possess a large specific surface area and a large pore size structure, which endows them with unique biological properties [32].

Due to the limitations of carbon NPs, such as a short half-life and rapid carbon monoxide release, leading to toxicity issues, MSNs are utilized as an alternative to enhance biological efficiency [63]. The large surface area and pore volume of MSNs, along with their excellent biocompatibility, make them effective therapeutic carriers. Studies have investigated whether mesoporous silica (MSN-A-CORM-2) encapsulated with carbon monoxide-releasing molecule-2 (CORM-2) enhances its anti-hypoxic effect in the cardiac cell lines H9 C2 (cardiomyoblasts) and 3 T3 (fibroblasts). CORM-2 is a compound designed to release controlled amounts of carbon monoxide, which is prepared primarily by binding to transition metals, organic small molecules, proenzymes, photosensitizing carriers, manganese, ruthenium, boron, and iron, for the prevention of ischemia, organ rejection, vascular dysfunction, and inflammation. Using scanning electron microscopy, transmission electron microscopy, and infrared spectroscopy, researchers have shown that encapsulation of hypoxic cardiac cells can significantly impact cell viability. Under hypoxic conditions, cell viability in H9 C2 and 3 T3 cells decreased by 46% and 54%, respectively. However, the encapsulation of CORM-2 in mesoporous silica increased cell viability by 30% in H9 C2 cells and 29% in 3 T3 cells among hypoxia-injured and H/R-injured cells, respectively [64]. The hypoxic environment at high altitudes stimulates sympathetic nerve activity and myocardial contractility, leading to increased hemodynamic load, which can induce acute cardiac decompensation, resulting in myocardial ischemia and potential myocardial infarction (MI) [65]. Additionally, hypoxia can stimulate excessive reactive oxygen species (ROS) production in mitochondria, impairing the body’s antioxidant capacity and disrupting the oxidative–antioxidant balance, potentially leading to lipid peroxidation and further affecting cardiac function [66]. Liu’s study utilized quercetin mesoporous silica nanoparticles (Q-MSNs) in a hypoxic environment on the myocardium of ischemia–reperfusion-injured rats. The study elucidated the mechanisms through the JAK2/STAT3 pathway, demonstrating that, compared to quercetin alone, Q-MSNs more effectively improved myocardial oxidative stress levels, inhibited myocardial cell apoptosis, reduced the infarcted area, and promoted cardiac blood flow recovery, thereby enhancing protective effects on the rat heart under hypoxic conditions [67].

### 4.3. Gold NPs in CMS

Gold was the first metallic element to be studied, and compared to other metals, it exhibits high resistance to oxidation and stability in chemical reactions [68]. Gold nanoparticles (AuNPs) are defined as gold particles with at least one dimension measuring less than 100 nm in a three-dimensional structure. Their basic units consist of tiny-sized particles, granting them unique physicochemical properties that macroscopic particles lack, such as optical properties, plasmonic resonance, fluorescence, and adsorption capabilities [33].

#### 4.3.1. Gold NPs in Pulmonary Arterial Hypertension

Studies indicate that lung inflammation plays a significant role in the formation and development of hypoxic pulmonary arterial hypertension (HPAH). One study found that hypoxia increased HIF-1α mRNA expression over time, peaking after 24 h, while HIF-2α expression also rose. However, NF-κB mRNA levels remained unchanged, and plasma levels of IL-1β and IL-6 stayed within the normal range. Prolonged exposure to hypoxia and low pressure resulted in a switch between HIF-1α and HIF-2α/NRF2, suggesting a close relationship between the development of HPAH, inflammation, and oxidative stress [69].

AuNPs exhibit strong anti-inflammatory and antioxidant capacities. Ponnani-kajamideen’s study demonstrated that AuNPs can eliminate lipid peroxidation and nitric oxide [70]. HPAH is characterized by airway hyperreactivity, which is associated with peribronchial fibrosis, mucus production, and the production of pro-inflammatory cytokines [71]. The effects of NPs on monocrotaline (MCT)-induced HPAH showed that NPs increased the activities of statin (SH), SOD, and NRF2 while decreasing oxidized glutathione (GSSG) levels. These changes significantly reduced the right ventricular hypertrophy index, indicating the alleviation of HPAH symptoms [8]. Free radicals play an important role in cellular responses under hypoxic conditions, with high concentrations leading to oxidative stress and cytotoxicity through the oxidation of nucleic acids, proteins, and lipids. Free radicals are primarily produced in the redox centers of the respiratory chain. Fan’s study utilized fluorescent nanodiamonds (FNDs) to detect free radicals and employed quantum sensing to explore the relationships between hypoxia, free radicals, and cellular redox states, contributing to future investigations of AuNPs and cellular oxidative stress [72].

#### 4.3.2. Gold NPs in Myocardial Injury

Based on the structure and unique electrophysiological properties of myocardial tissue, improving existing myocardial tissue engineering materials to better mimic the microenvironment required for excitation–contraction coupling has become a key focus and challenge in the field. Recent studies indicate that novel conductive nano-materials, with their excellent electrical conductivity, nanoscale size, and superior biocompatibility, can effectively enhance the electrical and mechanical microenvironment for seed cell survival. When combined with traditional natural or synthetic tissue engineering materials, these NPs can support engineered cardiac tissue (ECT) transplantation, facilitating good electrophysiological integration between ECT and the in vivo myocardium after transplantation [73].

An innovative nanoplatform provides the perfect strategy for developing nanoprobes for real-time in situ monitoring by combining metallic Se with AuNPs instead of the traditional gold–sulfur bond. This study simultaneously monitored the expression of myocardial apoptosis markers, Lon protease (Lon) and caspase-3, in cardiomyocytes under hypoxic conditions with high precision using the new AuNPs. Additionally, the nanoprobe was coupled with the mitochondrial H₂O₂ probe MitoPY1, demonstrating the relationship between altered reactive oxygen species (ROS) and myocardial apoptosis [74]. Apoptosis and oxidative stress are key factors contributing to organismal damage in high-altitude environments [75]. As an important transcription factor, HIF-1α plays a significant role in the metabolic mechanisms of hypoxia. AuNPs can also be functionalized using sulfhydryl chemistry with HIF-1α junctions suitable for biological applications, significantly increasing their stability. By leveraging the peroxidase-like properties of AuNPs, the expression of HIF-1α can be detected more sensitively in the myocardial tissues of infarcted rats in hypoxic conditions [76].

### 4.4. Magnetic NPs in CMS

Unlike most existing biomedical NPs, magnetic nanoparticles, such as iron oxide nanoparticles, can mediate external fields to produce local magnetic, thermal [77], and mechanical effects [78], as well as intrinsic enzyme-like catalytic activities [34]. Moreover, iron oxide nanoparticles are among the few inorganic NPs approved by the U.S. Food and Drug Administration (FDA) for clinical use.

Magnetic nanoparticles can generate mechanical effects in gradient or rotating magnetic fields, a phenomenon referred to as “nanomagnetism”. This can modulate cellular functions, such as disrupting the cell membrane or cytoskeleton, leading to apoptosis or necrosis. Shen et al. reported that magnetic nanoparticles combined with epidermal growth factor (EGF) can self-assemble under a low-frequency field to form a magnetic knife (MNPS-EDF), resulting in lysosomal membrane rupture and cellular necrosis. The formation of these magnetic knives led to the cleavage of lysosomal and cellular membranes, causing the death of more than 90% of the cells [79]. Furthermore, nanomagnetism can promote stem cell differentiation and the formation of functional tissues. Kang et al. demonstrated that by applying external fields of varying frequencies, they could control the oscillation rate of iron oxide nanoparticles complexed with spermacetyl-glycyl-aspartic acid (SPIO-RGD), facilitating remote manipulation of macrophage adhesion and polarization phenotypes in vivo and ex vivo [80]. Inorganic NPs can mitigate myocardial injury, restore cardiomyocyte function, and inhibit mitochondrial autophagy in hypoxic environments by modulating or silencing the expression of specific genes associated with HIF-1α and HIF-2α [69,80]. Therefore, further exploration of other inorganic nanoparticles is warranted to study pulmonary hypertension and myocardial injury in high-altitude environments.

Under hypoxic conditions, transcription factors activate the HIF-1α/basic fibroblast growth factor (bFGF) pathway, enhancing epithelial cell production and survival. HIF-1α is a key promoter of hypoxia-induced gene transcription and regulates angiogenesis through vascular endothelial growth factor (VEGF) and bFGF [81,82], both of which are crucial for promoting new blood vessel formation through adaptive transcriptional changes in response to hypoxia. The literature has shown that VEGF pre-treatment can increase post-infarction blood vessel levels, enhance blood vessel density, and mitigate myocardial injury [83]. Studies using superparamagnetic iron oxide (SPIO) nanoparticle-labeled adipose-derived stem cells (SPIOASCs) implanted in infarcted myocardium have shown that hypoxia culture increases HIF-1α and VEGF mRNA expression in both ASCs and SPIOASCs. The results confirm that interventions with hypoxia or SPIOASCs promote therapeutic angiogenesis and enhance cardiac function recovery in the infarcted myocardium [84]. Mesoporous iron oxide nanoparticles (MIONs) serve as novel nanoparticles for constructing a targeted release system for diallyl trisulfide (DATS) (DATS@MION-PEG-LF). This system extends the circulation time and targets the brain, exhibiting excellent biocompatibility, a controllable release pattern of H_2_S, and protective effects against in vitro hypoxia/reoxygenation after brain and cardiac ischemia. The protective effects primarily involve anti-apoptotic, anti-inflammatory, and antioxidant mechanisms [85].

Magnetic resonance imaging (MRI) faces limitations in accurate quantification and sensitivity. These limitations can be overcome by designing multimodal nanoprobes that integrate with other imaging modalities to improve diagnostic sensitivity, accuracy, and quantification [86]. For example, Li et al. developed Fe_3_O_4_/Au nanoparticles through a hydrothermal method and utilized them for magnetic resonance–computer tomography (MRI-CT) dual-modality imaging [87]. In another study by Hnatiuk, allogeneic bone marrow mesenchymal stem cells (BMMSCs) transfected with a microcyclic vector encoding mutant HIF1-a (BMMSC-HIF) were injected into myocardial-infarcted sheep, resulting in a 71.3% reduction in infarct volume. Enhanced long-term retention of BMMSC-HIF, attributed to HIF-1 overexpression, increased new vessel formation, and heart protection [88] was observed using paramagnetic iron nanoparticles for cellular tracking. Studies on CMR testing for CMS [14,48] suggest that NPs could be integrated with CMS assessments in the future.

## 5. Application of Organic NPs in CMS

In addition to inorganic NPs, various organic NPs are likewise used as carriers for delivering drugs used to treat diseases. Currently, there are different types of organic NPs used for therapeutic CMS, including liposomes, micelles, polylactic acid-hydroxyacetic acid (PLGA), and extracellular vesicles (ECs).

### 5.1. Liposomes

Liposomes, a type of tiny lipid vesicle, are increasingly popular nanocarriers featuring an aqueous core surrounded by one or more phospholipid bilayers. These layers mimic the structure of biological membranes, enhancing biocompatibility and biodegradability. Liposomes are advantageous for prolonging the half-life of drugs, controlling drug release, and improving biocompatibility and safety [89]. Additionally, they can deliver their payload selectively to disease sites through passive and/or active targeting, reducing systemic side effects, increasing tolerated doses, and improving therapeutic outcomes [90].

Currently, liposomal nanoparticles are employed in various medical settings. Treprostinil is commonly used in the treatment of HPAH, but it is not widely used in the clinic because of its short half-life and the systemic and local adverse effects caused by inhalation. Leifer’s study designed an inhaled sustained-release formulation of treprostinil (TRE), a common treatment for HPAH, utilizing lipid nanoparticles. These NPs, prepared through a proprietary process, consist of TPD, the hydrocarbon squalane, phospholipid DOPC, and pegylated lipid cholesterol/PEG 2000, resulting in a highly stable nanoparticle that can be readily adapted for large-scale production. The findings indicated that TRE plasma coated with carbon NPs could prolong pulmonary vascular relaxation under hypoxia compared to conventional TRE solutions [91]. Vaghasiya et al. developed a topical nano-spray gel (NSG) formulation combining liposomal heparin sodium and ibuprofen for treating frostbite in extremely cold, high-altitude conditions [92]. Research indicates that heparin aids in endothelial cell and capillary microcirculation repair [93]. This study demonstrated that heparin liposomes promote wound healing both in vitro (using a scratch test on fibroblasts) and in vivo (in Sprague Dawley rats) while also reducing inflammatory cytokines such as IL-4, IL-6, IL-10, and TNF-α in the bloodstream [92]. High-altitude hypobaric hypoxia can lead to pulmonary vasoconstriction and capillary leakage, increasing pulmonary arterial pressure and causing fluid accumulation in alveolar cavities, resulting in high-altitude pulmonary edema (HAPE) [19]. Pulmonary artery pressure (PAP) is a crucial physiological indicator, and systems that release medication in response to PAP changes can facilitate early medical intervention. Li’s study showed that multivesicular liposomes loaded with the calcium channel inhibitor albenesulfonic acid could effectively treat HAPE, with the drug being released into the vasculature within one hour of elevated pulmonary pressures, quickly reducing PAP [94]. Although many drugs can improve the quality of life for patients with HPAH, they have several drawbacks, including short half-lives, instability, and lack of specificity, which limit their suitability for widespread clinical application [95]. Intravenous prostacyclin agonists are effective in treating HPAH; however, they come with side effects and administration challenges. In a study involving a rat model of HPAH, the levels of IL-6, IL-1β, and TGF-α were reduced by the injection of liposomal NPs containing a prostacyclin agonist, as well as significant improvements in the percentage of intima-media thickness in the pulmonary vasculature and a notable decrease in right ventricular pressure compared to the control group. This demonstrates the potential application of NPs in hypoxic environments [96].

### 5.2. Extracellular Vesicles

Extracellular vesicles are membranous vesicles with a lipid bilayer structure, released spontaneously or under specific conditions by various cells, including immune cells, mesenchymal stem cells, epithelial cells, cancer cells, and other subcellular components. These vesicles primarily include exosomes from the endosomal pathway, microvesicles from membrane outgrowths and apoptotic bodies from apoptosis. Due to challenges in determining their origin post-release, they are typically classified by size, with those under 100 nm or 200 nm termed small extracellular vesicles [97,98].

In high-altitude environments, endothelial cells and platelets are susceptible to damage, contributing to microangiopathy. Olaf’s study, which analyzed platelet- and endothelial-derived extracellular vesicles in plasma samples from 39 mountaineers via flow cytometry, found significant elevations in CD31 neg CD42 blow/neg levels, indicating that extracellular vesicles serve as an early and sensitive marker of vascular damage [99]. After endothelial cell injury occurs, adverse cardiovascular responses such as vasoconstriction and thrombosis generate large amounts of ROS, causing endothelial cell oxidative stress injury and inducing MI [100]. Growth hormone-releasing hormone (GHRH) agonists promote the repair of damaged cardiac tissues and hold significant potential in the treatment of MI. Xiang et al. designed a ROS-sensitive amphiphilic polymer, poly (ethylene glycol)–polypropylene sulfide–poly (ethylene glycol) (PEGPPS-PEG), which was combined with @MR409 nanovesicles to observe the effect of these composite nanoparticles on ROS levels in MI cells under hypoxic conditions. The results indicated that the composite NPs could promote the recovery of cardiac function, providing a new approach to the treatment of MI in a hypoxic environment [101].

### 5.3. Polylactic Acid–Hydroxyacetic Acid (PLGA)

Polylactic acid–hydroxyacetic acid (PLGA) is a degradable polymer compound widely used in the pharmaceutical field and beyond. PLGA can encapsulate various organic or inorganic materials, including small-molecule drugs, vaccines, proteins, and metal particles. This versatility makes PLGA an ideal drug delivery system with potential for targeted applications [102,103]. Studies have shown that PLGA-encapsulated traditional Chinese medicine polysaccharides can effectively enhance the body’s immune response [104].

Liu et al. reported an inhalable platform designed to overcome hypoxia-induced immune changes and alveolar damage. This platform utilizes encapsulated PLGA NPs with macrophage apoptotic vesicle membranes (cMABs) [105]. cMABs are preloaded with nanocomplexes (NCs) containing superoxide dismutase (SOD)/catalase (CAT) and modified with pathology-responsive chains of the macrophage growth factor colony-stimulating factor (CSF), which exhibits a high affinity for alveolar epithelial cells (AECs). In a mouse model of hypoxic acute lung injury, these NCs can be efficiently translocated into mitochondria, thereby inhibiting inflammatory vesicle-mediated injury to AECs. This platform effectively targets hypoxia-induced immune changes and alveolar damage and could be applied to various hypoxia-induced inflammatory lung injuries. Another study encapsulated HIF-1α siRNA in PLGA NPs using a complex emulsion method, which helped reveal mechanisms of bone marrow mesenchymal stem cell trafficking and the advantages of PLGA NPs applied to a high-altitude environment [106].

The above experiments confirmed the feasibility of PLGA NPs for loading drugs in hypoxic environments. Akagi et al. investigated the efficacy and safety of intratracheal administration of the prostacyclin analog bezoprost (BPS), which was encapsulated in PLGA NPs for use in a hypoxic and MCT HPAH rat model, as well as in human HPAH pulmonary artery smooth muscle cells (PASMCs). Their findings indicated that a single administration of BPS NPs significantly reduced right ventricular pressure, right ventricular hypertrophy, and HPAH in the rats, improved survival rates, and did not cause inflammatory cell infiltration, hemorrhage, or fibrosis in the liver, kidney, spleen, or heart following the administration of BPS NPs [107].

### 5.4. Micelles

Micelles, a type of polymer-based nanoparticles, are nanoclusters with a shell/core structure formed by the self-assembly of amphiphilic block copolymers in aqueous solutions. Their hydrophobic core can encapsulate various hydrophobic anticancer drugs without altering their chemical structures, enhancing the stability and solubility of the drugs. Meanwhile, their hydrophilic shell and nanoscale size facilitate drug release and accumulation in vivo [108]. However, micellar carriers are susceptible to internal environmental factors, which can cause structural damage, premature drug release, and other problems. Micelle NPs are formed through the self-assembly of surfactants or amphiphilic polymers in water, and they are widely used as carriers for targeted drug delivery [109].

One study developed hypoxia-responsive nanomicelles loaded with carmustine, a typical antitumor drug that acts by damaging DNA at the guanine O-6 position. In this nanosystem, hyaluronic acid acts as a tumor-targeting ligand, binding to CD44 receptors overexpressed on tumor cells, enhancing the stability of the nanoparticles [110].

Tilianin-based nanomicelles covalently attach polyethylene glycol (PEG) to aryl-sulfide, forming amphiphilic diblock polymers that address Tilianin’s water insolubility. These nanomicelles are highly effective hydrogen peroxide scavengers and inhibit caspase-3 activity, thereby protecting cells from hypoxia-induced cytotoxicity. They also reduce levels of malondialdehyde (MDA), IL-1, and TNF-α and suppress the expression of apoptosis, TLR 4, and nuclear transcription factor NF-κB p65 in hypoxia model rats [111].

The mechanism of action of nanocarrier-carried drugs on CMS is summarized in Table 2.

### 5.5. Other

In addition to the common organic NPs mentioned above, there are other NPs that have been used as drug carriers to intervene in CMS, and they have also shown promising results.

Most heart diseases, such as arrhythmias, MI, and myocardial hypertrophy, are caused by the prolonged hypoxia of cardiomyocytes, primarily due to myocardial ischemia resulting from vascular obstruction caused by lipid deposits in the walls of coronary arteries. This leads to abnormal cardiac functioning [112]. A strong association exists between MI and hemorrhagic shock, with fluid resuscitation rapidly replenishing blood volume and improving systemic perfusion. However, the use of large volumes of fluids can cause damage to the organism, increasing the burden on the heart and leading to coagulation abnormalities, cellular edema, and mitochondrial dysfunction, which limits therapeutic efficacy [113]. TPP@PAMAM-MR (TPP-MR) is a novel cardiac resuscitation fluid with strong targeting properties for myocardial mitochondria. This new nanostructure, formed by wrapping L-malic acid with triphenylphosphine modified with polyamidamine (PAMAM), significantly reduces myocardial damage by increasing myocardial output and improving pulmonary artery pressure. Further studies showed that TPP-MR improved mitochondrial function in cardiomyocytes, reduced ROS levels in myocardial tissues, and suppressed the expression of iron death-associated proteins GPX4, ACSL4, and COX2, leading to enhanced antioxidant capacity [114]. Human cardiac regeneration is limited by the lack of studies on cellular models of hypoxic cardiac tissue. Nanofibrous structures are suitable for transplanting and culturing cells, as they can accurately mimic the three-dimensional structure of human cardiac tissue using various mats with the appropriate physicochemical properties. These properties include sufficient elasticity to allow for cell contraction, a high surface-area-to-volume ratio, and biocompatibility. Polyurethane nanofibrous mats have been shown to improve the expression of HIF-1α in human cardiomyocytes under hypoxic/reoxygenated conditions while simultaneously inhibiting cardiomyocyte apoptosis. Such studies support the search for an in vitro cardiac model and confirm the feasibility of using NPs to intervene in cardiomyocytes exposed to hypoxic environments [115]. Nitric oxide-releasing nanofibers have also been extensively investigated as potential therapeutic agents. In a study by Lee et al., biocompatible NO-releasing nanofibers were prepared from a mixture of polymers and methylaminopropyltrimethoxysilane (MAP 3) to assess their protective effect against HR injury in H9c2 cells. The results similarly demonstrated that NPs could protect the myocardium from HR injury [116].

Chinese medicines, known for their multi-component, multi-pathway, and multi-target properties, are safe and effective in treating HPAH. Combining them with NPs creates innovative drug formulations that overcome the limitations of current treatments. Research by Haddad et al. highlights the use of epigallocatechin gallate (EGCG), a key green tea component [117], in NP-based treatments for HPAH. EGCG is noted for its broad pharmacological benefits, including protection against cardiovascular and neurodegenerative diseases, diabetes, and cancer [118]. It also addresses iron accumulation and apoptotic injury in the hippocampus caused by high altitudes while enhancing NRF 2 and HO-1 expression to protect cardiac function [14,119].

NPs have been shown to act as antioxidants, providing cytoprotective and disease-modifying therapeutic effects by scavenging ROS and reducing cellular oxidative stress. They also serve as effective drug delivery platforms for cardiovascular protective medications, enhancing the short half-life of conventional drugs in vivo. Thus, scavenging ROS and eliminating oxidative stress are crucial in treating CMS. Several studies have investigated the relationship between NPs and ROS within the framework of nanoparticle drug delivery systems [31,47,66,74,120]. These investigations indicate that both inorganic and organic NPs can pose risks to organisms, with NPs potentially reaching pulmonary circulation and contributing to increased pulmonary artery pressure and vascular tension, which can lead to atherosclerosis and ischemia [66,120]. Currently, research on treating CMS through ROS reduction through NPs is limited and requires further exploration to advance this therapeutic strategy. We summarize the mechanisms by which different NPs act on CMS in Figure 3 and Table 3.

## 6. Conclusions and Perspectives

Inorganic NPs offer several advantages, including ease of preparation, controllable shape and size, and simple surface modification. Their unique optical, electrical, and magnetic properties provide potential applications in imaging, targeted delivery, and synergistic drug therapy. These properties can lead to prolonged therapeutic effects, reduced drug side effects, and improved therapeutic efficiency. This study reviews the recent research progress on NPs in the treatment of CMS. It first summarizes the current applications of inorganic and organic NPs in the treatment and diagnosis of CMS. These NPs can intervene in HPAH and myocardial injury through various mechanisms. However, due to the general toxicity, environmental impact, and high production costs of metallic materials, they remain in the stage of in vitro or animal testing, and their antioxidant biological mechanisms require further investigation.

NPs are used not only for treatment but also for the early diagnosis of CMS pathogenesis. CMR imaging is widely used as a non-invasive diagnostic tool for patients with CMS. It avoids the use of radiography or radioactive contrast agents, resulting in minimal side effects and a favorable safety profile. Numerous NPs can serve as MRI contrast agents, significantly enhancing the resolution and sensitivity of magnetic resonance imaging (MRI). These agents have been widely utilized in clinical diagnostics, including MRI nanoprobes that target endothelial cells and macrophages. Gadolinium complexes are the most commonly employed T1 contrast agents in clinical settings. However, the free gadolinium ions released during the metabolism of these contrast agents can be nephrotoxic and may lead to tissue fibrosis [121]. By contrast, magnetic iron oxide nanoparticles exhibit good biocompatibility and superparamagnetism, with a tendency to be phagocytosed by the liver, spleen, and monocyte–macrophage system [122]. This suggests that they accumulate in areas of inflammatory responses, which can be effectively identified. In the future, this technique could see increased application in diagnosing altitude-related illnesses.

NPs serve as gene carriers by encapsulating gene therapy molecules, such as DNA and RNA, or adsorbing them onto their surfaces. Specific targeting molecules, such as ligands and monoclonal antibodies, can be coupled to the surface of these nanoparticles. These targeting molecules bind to specific receptors on the surfaces of cells, facilitating cellular uptake and enabling safe and effective gene therapy. For instance, degradable nanoparticles can enhance the intracellular transport of VEGF antisense oligonucleotides into human retinal pigment epithelial cells, inhibiting mRNA expression and VEGF secretion. Polyamidoamine (PAMAM) dendrimers can serve as plasmid DNA carriers in place of β-cyclodextrin, with the PAMAM dendrimer/DNA complex enhancing catalase expression by 200-fold in in vitro assays [123]. Given their unique potential, nanoparticles are expected to see widespread use in gene therapy as nanotechnology continues to develop.

Gene therapy targets various diseases caused by genetic abnormalities at the nucleic acid level. This approach can introduce therapeutic genes to correct missing or mutated genes in situ or inhibit the abnormal expression of endogenous genes for therapeutic purposes. Nucleic acid drugs used in gene therapy include plasmid DNA (pDNA) that encodes specific proteins, messenger RNA (mRNA), antisense oligonucleotides for regulating target gene expression, short hairpin RNA, and small interfering RNA (siRNA). Among these, siRNAs developed based on the RNA interference (RNAi) mechanism have garnered significant attention due to their high specificity and efficiency. siRNAs can be encapsulated or conjugated with natural or synthetic carriers, facilitating efficient delivery to specific sites, cellular internalization in targeted tissues, and protection from serum degradation and macrophage phagocytosis [124]. These vectors encompass both viral and non-viral types. However, the use of viral vectors raises safety concerns, such as high immunogenicity and the risk of insertional mutagenesis, while their low drug-carrying capacity and high production costs limit clinical applications. Recently, researchers have increasingly favored safe cationic liposomes, polymers like polyethyleneimine (PEI), and inorganic NPs, which possess large gene-loading capacities and are amenable to scale-up. With the ongoing advancements in nanotechnology and gene therapy, their future use is anticipated to be widespread.

## Figures and Tables

**Figure 1 pharmaceutics-16-01375-f001:**
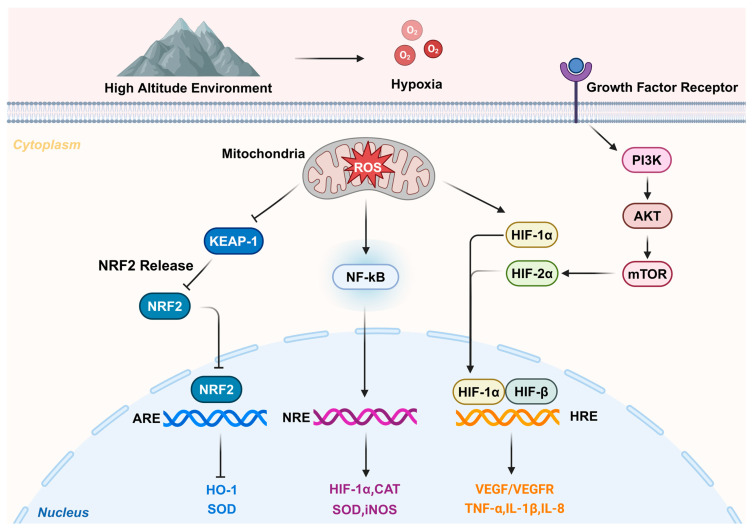
Possible functions and mechanisms in the pathogenesis of chronic mountain sickness. After plateau exposure occurs, the oxygen content of the air drops rapidly and oxidative stress occurs rapidly. KEAP-1 activation is inhibited, which in turn affects the expression of the antioxidant factor NRF2 and downstream factors. Meanwhile, the hypoxic environment leads to the occurrence of the inflammatory response and autophagy, activating the NF-kB and HIF signaling pathways to produce inflammatory molecules. ROS, reactive oxygen species; KEAP-1, Kelch-like ECH-associated protein 1; NRF2, nuclear factor erythroid 2-related factor 2; HO-1, heme oxygenase-1; SOD, superoxide dismutase; NF-κB, nuclear factor kappa-B; CAT, catalase; iNOS, inducible nitric oxide synthase; PI3K: phosphatidyl-inositol 3-kinase; AKT, serine–threonine kinase, mTOR, mammalian target of rapamycin; HIF-1α, hypoxia-inducible factor-1α; VEGF, vascular endothelial growth factor; TNF-α, tumor necrosis factor-α; IL-6, interleukin-6; IL-8, interleukin-8; ARE, anti-oxidative response element; NRE, negative regulatory element; HRE, hypoxia response element.

**Figure 2 pharmaceutics-16-01375-f002:**
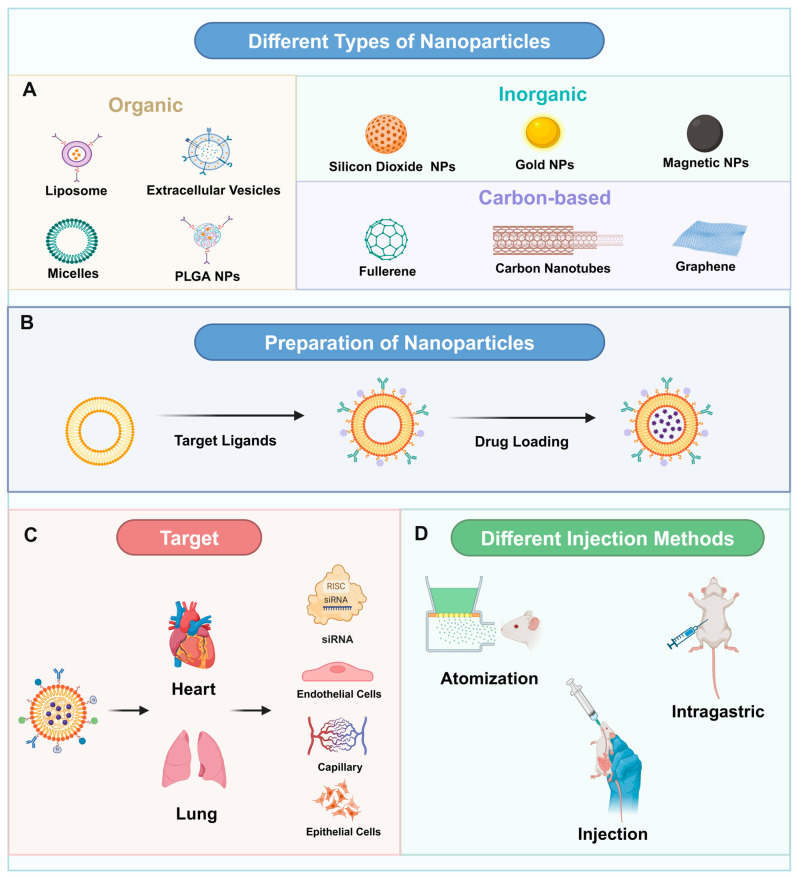
Preparation and delivery of nanoparticles. (**A**) Classification of organic and inorganic nanoparticles. (**B**) The preparation of nanoparticles mainly includes three steps: the selection of nanocarriers, the modification of surface molecules on nanocarriers, and agent encapsulation. (**C**) Nanoparticles affecting relevant targets in patients with chronic mountain sickness. (**D**) Nanoparticles are delivered to the target site by different injection modes, including intraperitoneal injection, oral administration, and nebulized administration.

**Figure 3 pharmaceutics-16-01375-f003:**
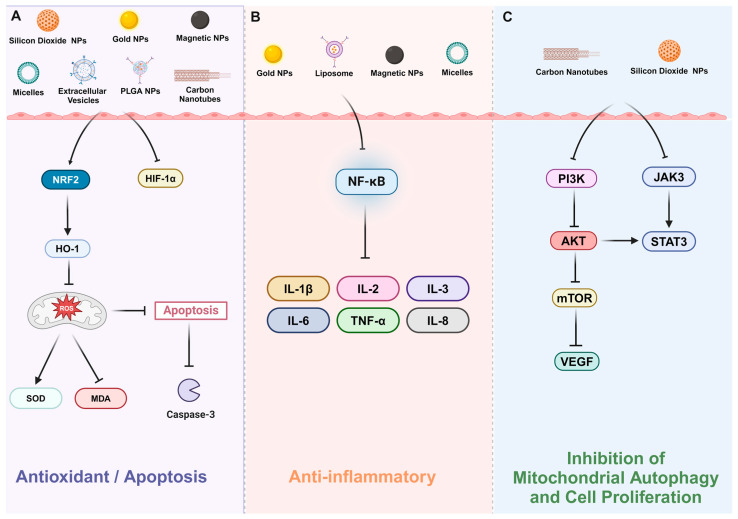
Different types of nanoparticles can intervene in chronic mountain sickness through different pathways. After CMS occurs, oxidative stress is rapidly activated due to hypoxia, and inflammatory responses and autophagy happen simultaneously. (**A**) Nanoparticle-based drug delivery systems can elevate the NRF2-HO-1 pathway activity to inhibit the production of ROS and suppress apoptosis. (**B**) Nanoparticle-based drug delivery systems can regulate the NF-κB pathway to reduce inflammatory responses. (**C**) Nanoparticle-based drug delivery systems can also inhibit mitochondrial autophagy hypoxia-induced cell proliferation via PI3K/AKT and JAK3/STAT3 pathways. AKT, serine–threonine kinase; Caspase-3, cysteinyl aspartate specific proteinase-3; CAT, catalase; HIF-1α, hypoxia-inducible factor-1α; HO-1, heme oxygenase-1; IL-6, interleukin- 6; IL-8, interleukin-8; JAK3, Janus Kinase 3; MDA, malondialdehyde; mTOR, mammalian target of rapamycin; NF-κB, nuclear factor kappa-B; NRF2, nuclear factor erythroid 2-related factor 2; PI3K: phosphatidyl-inositol 3-kinase; ROS, reactive oxygen species; SOD, superoxide dismutase; STAT3, signal transduction and activator of transcription 3; TNF-α, tumor necrosis factor-α; VEGF, vascular endothelial growth factor.

**Table 2 pharmaceutics-16-01375-t002:** Target molecules with their targets, properties, and carriers in chronic mountain sickness.

No.	Ligands	Targets	Properties	NPs	Mechanism of Action	References
1	Heparin sodium	Endothelial cells and capillaries	High stability and quick access to the target	Liposomes	Modulation of the inflammatory cytokine	[92]
2	Amlodipine besylate	Pulmonary capillaries	Hydrostatic pressure sensitivity and ability to carry multiple drugs	Pressure-sensitive multivesicular liposomes	Targeted treatment of HAPE using hydrostatic-pressure-sensitive multivesicular liposomes with simultaneous inhibition of HIF-1α expression	[94]
3	ICAM-1, VCAM-1, and VE-cadherin	Platelets and endothelial cells	Early reflection of endothelial function damage from hypoxia	Extracellular vesicles	Endothelial cell apoptosis and coagulation, as reflected by CD 31 and CD42b expression, and inflammatory immune response, as measured by ICAM-1, VCAM-1, and VE-Cadherin	[99]
4	Superoxide dismutase/catalase nanocomplexes	Alveolar epithelial cells	Altered macrophage phenotype and high affinity for targets	Camouflaged PLGA microparticles with macrophage apoptotic body membrane	Alteration of the phenotype of circulating monocytes and macrophages to resolve lung injury and inflammation	[105]
5	Plasmid and lentivirus	Mesenchymal stem cell and retinal pigment epithelial cells	Creation of a bionic delivery system that is highly targeted	PLGA	Inhibition of HIF-1α mRNA in epithelial cells	[106]
6	Polyethylene glycol compound attached to propylene sulfide formed amphiphilic diblock polymer	Cardiomyocytes	Highly efficient hydrogen peroxide scavengers	Tilianin-loaded micelles	Inhibition of myocardial injury by reducing inflammation, oxidative stress, and apoptosis	[111]

PLGA: polylactic acid–hydroxyacetic acid; HAPE: high-altitude pulmonary edema.

**Table 3 pharmaceutics-16-01375-t003:** Relevant pathways and carriers for NP therapy.

No.	Drug	NPs	Pathway	Effect	References
1	Methotrexate	Lipid core nanoparticles	ROS-VEGF	MTX-LDE increased antioxidant enzymes and reduced apoptosis, macrophages, ROS production, and hypoxic damage to myocardium.	[31]
2	Polyaniline	Nanofibrous polycaprolactone mats	Caspase-3–Bcl-2	Nanofibrous mats polymerized in situ with polyaniline reduce intracellular ROS content and caspase-3 mRNA expression and attenuate the hypertrophic effect of hypoxia on H9 c2 cells.	[55]
3	Carbon monoxide-releasing molecule-2	Mesoporous silica	Scanning electron microscopy and cell viability assay	Mesoporous silica NPs may lead to sustained release of CO and thus hypoxia/reoxygenation, resulting in minimization of toxic effects.	[64]
4	Quercetin	Mesoporous silica nanoparticles	JAK2/STAT3	Q-MSNs elevated JAK 2 and STAT 3 protein expression, decreased Bax, caspase-3, Bim, and Bid protein expression, and ameliorated cardiomyocyte apoptosis, myocardial infarction, and ventricular remodeling in hypoxic environment.	[67]
5	Heparin	Liposomes	IL-6-TNF-α	Frostbite was significantly improved in rats after aerosolized administration of nano-spray gel, with reductions in IL-6, TNF-α, IL-10, and IL-4.	[92]
6	Prostacyclin	Liposomes	IL-6, IL-1β, TGFβ	Prostacyclin agonist NPs improved clinical outcomes in HPAH after administration of elevated HGF expression and significant reductions in IL-6, IL-1β, and TGFβ.	[96]
7	Epigallocatechin gallate (EGCG)	Liposomes	TGFβ	Inhalable EGCG nanoliposomes have the function of inhibiting TGFβ signaling and could be potential treatment for HPAH.	[117]
8	Growth hormone-releasing hormone	nano PEG-PPS-PEG@MR409 vesicles	ROS	NPs encapsulating growth hormone-releasing hormone attenuate ROS content and apoptosis in posthypoxic myocardial infarction cells and restore cardiac function.	[101]
9	Acetated Ringer’s (AR) and Lactate Ringer’s solution (LR)	TPP@PAMAM-MR (TPP-MR)	Glutathione peroxidase 4	TPP@PAMAM-MR novel nanocrystal resuscitation solution improves cardiac and mitochondrial function in hypoxia-treated cardiomyocytes, attenuates ROS production, and inhibits iron-toxicity-associated GPX 4, ACSL 4, and COX 2 protein expression.	[114]

MTX: methotrexate; LDE: lipid core nanoparticles; ROS: reactive oxygen species; VEGF: vascular endothelial growth factor; Q-MSNs: quercetin-loaded mesoporous silica nanoparticles; IL-1β: interleukin-1β; IL-6: interleukin-6; IL-10: interleukin-10; TNF-α: tumor necrosis factor-α; TGFβ: transforming growth factor-β; HPAH: hypoxic pulmonary arterial hypertension.

## Data Availability

Data are contained within the article.

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
