# Peer review of "Research Progress on Using Nanoparticles to Enhance the Efficacy of Drug Therapy for Chronic Mountain Sickness"

_pharmaceutics, 2024, doi:10.3390/pharmaceutics16111375_

Round 1
Reviewer 1 Report
Comments and Suggestions for Authors
The authors present a well-crafted review on the proposed topic. The topic is very interesting and represents an example of an even wider scientific field in which pharmaceutical drugs are combined with inorganic nanoparticles. The vast amount of information is properly digested and explained through adequate schemes and tables. The manuscript should be published after relatively minor modifications.
1) Carbon based nanoparticles do not appear to include graphene oxide. Is there a reason for that?
2) Metallic nanoparticles because of their size and surface properties exhibit intrinsic catalytic properties. How do these properties interfere positively or negatively with the targeted biological pathways? For instance, Ni may catalyze the oxygen reduction reaction and may help to reduce ROS in purely chemical processes. At the same time, the use of Ni is hampered by its biological toxicity (see Table 1). This statement, by the way, is a bit of a euphemism because Ni is indeed carcinogenic.
Author Response
Dear Reviewer,
Thank you for giving us the opportunity to amend our manuscript and respond to the peer-review. According to your comments, we have modified our manuscript. Below are our detailed and point-by-point responses, which are marked in red font in the revised manuscript for easy viewing. We hope that the revisions and our responses will be sufficient to make our manuscript suitable for publication in Pharmaceutics and thank you again for your review.
Yours sincerely,
Dongdong Qin
Yunnan University of Chinese Medicine
Point-by-point responses
The authors present a well-crafted review on the proposed topic. The topic is very interesting and represents an example of an even wider scientific field in which pharmaceutical drugs are combined with inorganic nanoparticles. The vast amount of information is properly digested and explained through adequate schemes and tables. The manuscript should be published after relatively minor modifications.
- Carbon based nanoparticles do not appear to include graphene oxide. Is there a reason for that?
Response: We would like to thank the reviewer for this suggestion. After reviewing the published articles so far, we did not find any studies of graphene oxide interfering with CMS, which we have described in the manuscript (page x, lines x). Not only that, we have researched the literature and found that fullerenes have been used to intervene CMS, which we have added into the manuscript (page x, lines 268-277).
Page 9, lines 268-277: However, a search of the relevant literature did not reveal the use of graphene to intervene in CMS.
Page x, lines 226-234: Fullerene was discovered in 1985, which was the third allotrope of carbon after the discovery of graphite and diamond. Because of its spherical three-dimensional spatial structure, it can cause a lot of physical and chemical reactions. The study of fullerenes has been the general concern of the majority of scientific research workers, which has good antioxidant properties and is very effective for free radical scavenging [46]. Fullerene-C60-based particles have been shown to treat hypoxia-induced mitochondrial dysfunction in mammalian myocardium, and pharmacokinetics and pharmacodynamics indicate that it is suitable for administration in single or multiple injections for the prevention and treatment of clinical conditions involving acute and chronic myocardial hypoxia without acute toxicity [47].
- Metallic nanoparticles because of their size and surface properties exhibit intrinsic catalytic properties. How do these properties interfere positively or negatively with the targeted biological pathways? For instance, Ni may catalyze the oxygen reduction reaction and may help to reduce ROS in purely chemical processes. At the same time, the use of Ni is hampered by its biological toxicity (see Table 1). This statement, by the way, is a bit of a euphemism because Ni is indeed carcinogenic.
Response: We appreciate your suggestions. Ni is indeed carcinogenic and biological toxicity. After searching the relevant literature, we have found that there is no literature exploring how to intervene in CMS while eliminating the toxicity of Ni, so we deleted the use of Ni nanoparticle in Table 1.

Reviewer 2 Report
Comments and Suggestions for Authors
In this review article, authors report on the research progress on using inorganic nanomaterials to enhance the efficacy of drug therapy for chronic mountain sickness. This is an interesting review, well written and includes a lot of inorganic materials that have been used over the years for that purpose. I only have a few minor comments that could improve further the quality of this manuscript.
1. The are a few typos in the manuscript that need to be corrected. Page 3, line 105 please correct the name Saradas. Be consistent and add the reference numbers before the end of the sentence, for example on page 3 line 120 and line 125, the reference number is not added the same way. On page 8 line 242, the surname MORIMOTO doesn’t have to be in capital letters. Page 11, line 428, add the numbers in iron oxide as subscripts. These are a few examples, please check the manuscript for typos.
2. Please remove section 9 patents from page 21, it is not relevant to this manuscript.
3. For the keywords suggested, I would change nanomaterials to inorganic nanomaterials and use drug delivery separately.
4. In the introduction on page 2, line 49 I suggest mentioning e few examples of nanoparticles and on the same page line 53, a few example of inorganic nanocarriers. It is a bit confusing like that.
5. In general, I would ask permission to use more figures in the manuscript, to make it more interesting, it is difficult to follow some pathways without a figure.
Author Response
Dear Reviewer,
Thank you for giving us the opportunity to amend our manuscript and respond to the peer-review. According to your comments, we have modified our manuscript. Below are our detailed and point-by-point responses, which are marked in blue font in the revised manuscript for easy viewing. We hope that the revisions and our responses will be sufficient to make our manuscript suitable for publication in Pharmaceutics and thank you again for your review.
Yours sincerely,
Dongdong Qin
Yunnan University of Chinese Medicine
Point-by-point responses
In this review article, authors report on the research progress on using inorganic nanomaterials to enhance the efficacy of drug therapy for chronic mountain sickness. This is an interesting review, well written and includes a lot of inorganic materials that have been used over the years for that purpose. I only have a few minor comments that could improve further the quality of this manuscript.
- The are a few typos in the manuscript that need to be corrected. Page 3, line 105 please correct the name Saradas. Be consistent and add the reference numbers before the end of the sentence, for example on page 3 line 120 and line 125, the reference number is not added the same way. On page 8 line 242, the surname MORIMOTO doesn’t have to be in capital letters. Page 11, line 428, add the numbers in iron oxide as subscripts. These are a few examples, please check the manuscript for typos.
Response: Thank you for reviewing our manuscript very carefully. We overlooked these important details, and we have corrected them in the revised manuscript.
(1) We have corrected the name Saradas as Sarada S (page 3, lines 105-107).
Page 3, lines 105-107: Sarada S et al. observed significantly increased levels of ROS, lipid peroxidation, and superoxide dismutase (SOD) in rats exposed to a low-pressure hypoxic environment at an altitude of 7620 m.
(2) We have added the reference numbers before the end of the sentence (page x, lines x).
Page 3, lines 123-124: This suggests that IL-6 is more sensitive to high-altitude exposure [21].
(3) We have corrected the surname MORIMOTO as Morimoto and checked the entire manuscript for similar mistakes (page 9, lines 250-251).
Page 9, lines 250-251: Morimoto's study confirmed that exposure to fullerenes does not induce inflammation or causes only transient inflammation in the lungs.
(4) We have added the numbers in iron oxide as subscripts and checked throughout the entire manuscript for the same similar blunders (page 13, lines 455-457).
Page 13, lines 455-457: Li et al. developed Fe3O4/Au nanoparticles through a hydrothermal method and utilized them for magnetic resonance-computer tomography (MRI-CT) dual-modality imaging [88].
- Please remove section 9 patents from page 21, it is not relevant to this manuscript.
Response: Thanks for your concern. We have identified the problem at the time of submission and have made the corresponding deletions.
- For the keywords suggested, I would change nanomaterials to inorganic nanomaterials and use drug delivery separately.
Response: Your suggestions are very insightful. By combining your and other reviewers' comments, our manuscript not only discusses inorganic nanoparticles, but also introduces organic nanoparticles. Therefore, we have retitled our manuscript as “Research Progress on Using Nanoparticles to Enhance the Efficacy of Drug Therapy for Chronic Mountain Sickness”. As you suggested, it is suitable that “drug delivery” should be included in the keywords.
- In the introduction on page 2, line 49 I suggest mentioning a few examples of nanoparticles and on the same page line 53, a few example of inorganic nanocarriers. It is a bit confusing like that.
Response: Thanks again for your constructive suggestions to our manuscript. We have provided the examples in the introduction to make the logic clearer. (page 2, lines 53-59).
Page 2, lines 53-59: Inorganic NPs, as a novel class of drug carriers, show promising potential in achieving targeted drug delivery, controlling drug release, and improving the solubility and bioavailability of poorly soluble drugs, thereby enhancing drug efficacy. For example, non-metallic materials include mesoporous silica and carbon NPs and metallic materials consist of silver, gold, iron oxide and nickel oxide NPs [7]. Organic NPs include liposomes, extracellular vesicles, polylactic acid-hydroxyacetic acid (PLGA), and micelles.
- In general, I would ask permission to use more figures in the manuscript, to make it more interesting, it is difficult to follow some pathways without a figure.
Response: Thanks for your valuable comments. We have used more figures to make our manuscript more interesting, especially providing the involved molecular pathways, as illustrated in Figure 3.
Figure 3. Different types of nanoparticles can intervene in chronic mountain sickness through different pathways. After CMS occurs, oxidative stress is rapidly activated due to hypoxia, and inflammatory responses and autophagy happen simultaneously. (A) Nanoparticle-based drug delivery systems can elevate the NRF2-HO-1 pathway to inhibit the production of ROS, and suppress the apoptosis. (B) Nanoparticle-based drug delivery systems can regulate the NF-κB pathway to reduce the inflammatory responses. (C) Nanoparticle-based drug delivery systems can also inhibit mitochondrial autophagy hypoxia-induced cell proliferation via PI3K/AKT and JAK3/STAT3 pathways. AKT, Serine-threonine kinase; Caspase-3, Cysteinyl aspartate specific proteinase-3; CAT, Catalase; HIF-1α, Hypoxia-inducible factor-1α; HO-1, Heme oxygenase-1; IL-6, Interleukin- 6; IL-8, Interleukin-8; JAK3, Janus Kinase 3; MDA, Malondialdehyde; mTOR, Mammalian target of rapamycin; NF-κB, nuclear factor kappa-B;NRF2, Nuclear factor erythroid 2-related factor 2; PI3K: Phosphatidyl-inositol 3-kinase; ROS, Reactive Oxygen Species; SOD, Superoxide dismutase; STAT3, Signal transduction and activator of transcription 3; TNF-α, Tumor Necrosis Factor-α; VEGF, Vascular endothelial growth factor.

Reviewer 3 Report
Comments and Suggestions for Authors
The review “Research Progress on Using Inorganic Nanomaterials to Enhance the Efficacy of Drug Therapy for Chronic Mountain Sickness” describes the pathogenesis, symptoms and physiological modification associated with Chronic Mountain Sickness (CMS) and provide some data concerning the potential use of inorganic materials for its treatment. The data are described more from medical and biochemical point of view, some data are duplicated and some have only a limited connection with the paper topic. Moreover, the authors claim that this condition can be simply treated by people’s relocation in lower elevations so I don’t understand why is need to use inorganic or hybrid nanoparticles (NPs) that finally leads also to unwanted side effects. The conclusions and perspectives are not consistent with data presented and with proposed Title. My overall comment is that these data present only a low interest for the readers interested in developing some effective strategies for the therapy of this condition. I therefore cannot recommend this paper for publication having in view the following aspects:
- Data are not systematic and clear presented.
- All species described in Figure 1A are NPs and not nanomaterial and Carbon and Magnetic are ambiguous terms for the species described under these names. Moreover, Nickel is not the system tested but its oxide and moreover this system proved only the toxicity and not beneficial action for humans.
- The authors make a confusion between an inorganic nanomaterial and a nanocarriers (Fig. 1 B).
- Some properties described at nanomaterials are not connected with their biological applications.
- Not all carbon nanomaterials described at 4.1 were tested for CMS potential applications.
- All species described at 4 as inorganic materials are not of this type but hybrid ones. The properties of NPs are described at each subchapter sometimes with the same words.
- The composition of some species is uncertain (i.e. what is the composition of Au-Se or iron NPs, carbon monoxide-releasing molecule, peroxidase nanocomplexes, ONO 1301???).
- The NiO NPs exhibit only toxicity so subchapter 4.5 is not suitable for proposed topic.
- All nanocarriers described in chapter 5 are organic based species and some have only a limited connection with proposed topic (i.e. 5.2, information concerning macrophages, gene therapy at the end).
- The nanocarriers described in Chapter 5 and Nanomaterials from Chapter 6 are similar species.
- In Table 2 is not presented the mechanism of action of nanocarrier-carried drugs!
- The data provided in Chapter 7 not represent Future Research Advances in Nanomaterials for CMS.
- The references are careless presented.
Author Response
Dear Reviewer,
Thank you for giving us the opportunity to amend our manuscript and respond to the peer-review. According to your comments, we have modified our manuscript. Below are our detailed and point-by-point responses, which are marked in green font in the revised manuscript for easy viewing. We hope that the revisions and our responses will be sufficient to make our manuscript suitable for publication in Pharmaceutics and thank you again for your review.
Yours sincerely,
Dongdong Qin
Yunnan University of Chinese Medicine
Point-by-point responses
The review “Research Progress on Using Inorganic Nanomaterials to Enhance the Efficacy of Drug Therapy for Chronic Mountain Sickness” describes the pathogenesis, symptoms and physiological modification associated with Chronic Mountain Sickness (CMS) and provide some data concerning the potential use of inorganic materials for its treatment. The data are described more from medical and biochemical point of view, some data are duplicated and some have only a limited connection with the paper topic. Moreover, the authors claim that this condition can be simply treated by people’s relocation in lower elevations so I don’t understand why is need to use inorganic or hybrid nanoparticles (NPs) that finally leads also to unwanted side effects. The conclusions and perspectives are not consistent with data presented and with proposed Title. My overall comment is that these data present only a low interest for the readers interested in developing some effective strategies for the therapy of this condition. I therefore cannot recommend this paper for publication having in view the following aspects:
- Data are not systematic and clear presented.
Response: Thank you for your comments. By combining your and other reviewers' comments, we have reorganized the manuscript to ensure the data are systematically and clearly presented.
- All species described in Figure 1A are NPs and not nanomaterial and Carbon and Magnetic are ambiguous terms for the species described under these names. Moreover, Nickel is not the system tested but its oxide and moreover this system proved only the toxicity and not beneficial action for humans.
Response: We would like to thank you for the valuable comments. We did confuse the concepts of nanomaterials and nanoparticles before, and as you suggested, we have replaced the nanomaterials with nanoparticles. Consistent with other reviewers’ comments, Nickel is indeed carcinogenic and biological toxicity. After searching the relevant literature, we found that there were no studies exploring how to intervene in CMS while eliminating the toxicity of Nickel, and therefore we deleted the description of Nickel.
- The authors make a confusion between an inorganic nanomaterial and a nanocarriers (Fig. 1 B).
Response: We greatly appreciate your valuable comments. We have replaced the nanomaterials with nanoparticles.
- Some properties described at nanomaterials are not connected with their biological applications.
Response: Thank you very much for your concerns. In Table 1, we have added a column of “Biological application” to make the properties more connected with biological applications.
5.Not all carbon nanomaterials described at 4.1 were tested for CMS potential applications.
Response: Thank you for your invaluable comments. As you said, not all carbon nanomaterials described at 4.1 were tested for CMS potential applications. Following your suggestion, we have provided an argument in the revised manuscript (page 10, lines 311-312).
Page 10, lines 311-312: However, a search of the relevant literature did not reveal the use of graphene to intervene in CMS.
- All species described at 4 as inorganic materials are not of this type but hybrid ones. The properties of NPs are described at each subchapter sometimes with the same words.
Response: We deeply appreciate your valuable comments. Nanoparticles developed for nanomedicine could be classified into three main categories: 1) inorganic, 2) organic and 3) organic–inorganic hybrids. Inorganic nanoparticles include non-metal (such as quantum dots, carbon nanoparticles, mesoporous silica nanoparticles) and metal nanoparticles, including gold, silver, iron oxide, cerium oxide, etc. Organic NPs include liposomes, extracellular vesicles, polylactic acid-hydroxyacetic acid (PLGA), and micelles, etc. Up until now, there are no studies using organic–inorganic hybrids to treat CMS.
- The composition of some species is uncertain (i.e. what is the composition of Au-Se or iron NPs, carbon monoxide-releasing molecule, peroxidase nanocomplexes, ONO 1301???).
Response: Thank you for your comments. It is true that we did not describe the composition of some species clearly in the previous manuscript. In the revised manuscript, we have added them (pages 14-15, lines 546-550).
Pages 14-15, lines 546-550: This platform utilizes encapsulated PLGA NPs with macrophage apoptotic vesicle membranes (cMAB) [106]. cMAB is preloaded with nanocomplexes (NC) containing superoxide dismutase (SOD)/ catalase (CAT) and modified with pathology-responsive chains of the macrophage growth factor colony-stimulating factor (CSF), which exhibits a high affinity for alveolar epithelial cells (AECs)
Page 10, lines 328-332: CORM-2 is a compound designed to release controlled amounts of carbon monoxide, which is prepared primarily by binding to transition metals, organic small molecules, proenzymes, photosensitizing carriers, manganese, ruthenium, boron, and iron, for the prevention of ischemia, organ rejection, vascular dysfunction, and inflammation.
Page 14, lines 503-512: Although many drugs can improve the quality of life for patients with HPAH, they have several drawbacks, including short half-lives, instability, and lack of specificity, which limit their suitability for widespread clinical application [96]. Intravenous prostacyclin agonists are effective in treating HPAH; however, they come with side effects and administration challenges. In a study involving a rat model of HPAH, levels of IL-6, IL-1β, and TGF-α were reduced by injection of a liposomal NPs containing a prostacyclin agonist, as well as significant improvements in the percentage of intima-media thickness in the pulmonary vasculature and a notable decrease in right ventricular pressure compared to the control group. This demonstrates the potential application of NPs in hypoxic environments [97].
- The NiO NPs exhibit only toxicity so subchapter 4.5 is not suitable for proposed topic.
Response: Thank you for your comments. Following your suggestion, we have deleted this subchapter.
- All nanocarriers described in chapter 5 are organic based species and some have only a limited connection with proposed topic (i.e. 5.2, information concerning macrophages, gene therapy at the end).
Response: Thank you for your professional comments. We did include some irrelevant information and in the revised manuscript, we have removed these parts that are not related to our topic.
- The nanocarriers described in Chapter 5 and Nanomaterials from Chapter 6 are similar species.
Response: Thank you for your comments. Our previous manuscript is not well organized. In the revised manuscript, we have combined these two chapters and discussed them according to different kinds of nanoparticles (page 14, lines 503-512 and pages 15-16 lines 589-644.
Page 14, lines 503-512: Although many drugs can improve the quality of life for patients with HPAH, they have several drawbacks, including short half-lives, instability, and lack of specificity, which limit their suitability for widespread clinical application [96]. Intravenous prostacyclin agonists are effective in treating HPAH; however, they come with side effects and administration challenges. In a study involving a rat model of HPAH, levels of IL-6, IL-1β, and TNF-α were reduced by injection of liposomal NPs containing a prostacyclin agonist, as well as significant improvements in the percentage of intima-media thickness in the pulmonary vasculature and a notable decrease in right ventricular pressure compared to the control group. This demonstrates the potential application of NPs in hypoxic environments [97].
Page 15-16, lines 589-644: 5.5 Other
In addition to the common organic NPs mentioned above, there are other NPs that have been used as drug carriers to intervene in CMS, and they have also shown promising results.
Most heart diseases, such as arrhythmias, MI, and myocardial hypertrophy, are caused by prolonged hypoxia of cardiomyocytes, primarily due to myocardial ischemia resulting from vascular obstruction caused by lipid deposits in the walls of coronary arteries. This leads to abnormal cardiac functioning [113]. A strong association exists between MI and hemorrhagic shock, with fluid resuscitation rapidly replenishing blood volume and improving systemic perfusion. However, the use of large volumes of fluids can be damaged to the organism, increasing the burden on the heart and leading to coagulation abnormalities, cellular edema, and mitochondrial dysfunction, which limits therapeutic efficacy [114]. TPP@PAMAM-MR (TPP-MR) is a novel cardiac resuscitation fluid with strong targeting properties for myocardial mitochondria. This new nanostructure, formed by wrap-ping L-malic acid with triphenylphosphine modified with polyamidamine (PAMAM), significantly reduces myocardial damage by increasing myocardial output and improving pulmonary artery pressure. Further studies showed that TPP-MR improved mitochondrial function in cardiomyocytes, reduced ROS levels in myocardial tissues, and suppressed the expression of iron death-associated proteins GPX4, ACSL4, and COX2, leading to enhanced antioxidant capacity [115]. Human cardiac regeneration is limited by the lack of studies on cellular models of hypoxic cardiac tissue. Nanofibrous structures are suitable for transplanting and culturing cells, as they can accurately mimic the three-dimensional structure of human cardiac tissue using various mats with the appropriate physicochemical properties. These properties include sufficient elasticity to allow for cell contraction, a high surface area-to-volume ratio, and biocompatibility. Polyurethane nanofibrous mats have been shown to improve the expression of HIF-1α in human cardiomyocytes under hypoxic/reoxygenated conditions while simultaneously inhibiting cardiomyocyte apoptosis. Such studies support the search for an in vitro cardiac model and confirm the feasibility of NPs to intervene in cardiomyocytes exposed to hypoxic environments [116]. Nitric oxide-releasing nanofibers have also been extensively investigated as potential therapeutic agents. In a study by Lee et al, biocompatible NO-releasing nanofibers were prepared from a mixture of polymers and methylaminopropyltrimethoxysilane (MAP 3) to assess their protective effect against HR injury in H9c2 cells. The results similarly demonstrated that NPs could protect the myocardium from HR injury [117].
Chinese medicines, known for their multi-component, multi-pathway, and multi-target properties, are safe and effective in treating HPAH. Combining them with NPs creates innovative drug formulations that overcome the limitations of current treatments. Research by Haddad et al. highlights the use of epigallocatechin gallate (EGCG), a key green tea component [118], in NP-based treatments for HPAH. EGCG is noted for its broad pharmacological benefits, including protection against cardiovascular, neurodegenerative diseases, diabetes, and cancer [119]. It also addresses iron accumulation and apoptotic injury in the hippocampus caused by high altitudes, while enhancing NRF 2 and HO-1 expression to protect cardiac function [89,120].
NPs have been shown to act as antioxidants, providing cytoprotective and disease modifying therapeutic effects by scavenging ROS and reducing cellular oxidative stress. They also serve as effective drug delivery platforms for cardiovascular protective medications, enhancing the short half-life of conventional drugs in vivo. Thus, scavenging ROS and eliminating oxidative stress are crucial in treating CMS. Several studies have investigated the relationship between NPs and ROS within the frame-work of nanoparticles drug delivery systems [43,48,64,73,121]. These investigations indicate that both inorganic and organic NPs can pose risks to organisms, with NPs potentially reaching pulmonary circulation and contributing to increased pulmonary artery pressure and vascular tension, which can lead to atherosclerosis and ischemia [64,121]. Currently, research on treating CMS through ROS reduction through NPs is limited and requires further exploration to advance this therapeutic strategy. We summarize the mechanisms by which different NPs act on CMS in Figure 3 and Table 3.
- In Table 2 is not presented the mechanism of action of nanocarrier-carried drugs!
Response: Thank you for your concerns. Following your suggestion, we have added a column of “Mechanism of action” in Table 2.
- The data provided in Chapter 7 not represent Future Research Advances in Nanomaterials for CMS.
Response: Thank you for this valuable suggestion. As your said, previous Chapter 7 is really not representative of future research progress. In the revised manuscript, we have rewritten this section (page 18, lines 668-681).
Page 18, lines 668-681: NPs are not only used for treatment, but also for early diagnosis of CMS pathogenesis. CMR imaging is widely used as a non-invasive diagnostic tool for patients with CMS. It avoids the use of radiography or radioactive contrast agents, resulting in minimal side effects and a favorable safety profile. Numerous NPs can serve as MRI contrast agents, significantly enhancing the resolution and sensitivity of magnetic resonance imaging (MRI). These agents have been widely utilized in clinical diagnostics, including MRI nanoprobes that target endothelial cells and macrophages. Gadolinium complexes are the most commonly employed T1 contrast agents in clinical settings. However, the free gadolinium ions released during the metabolism of these contrast agents can be nephrotoxic and may lead to tissue fibrosis [122]. By contrast, magnetic iron oxide nanoparticles exhibit good biocompatibility and superparamagnetism, with a tendency to be phagocytosed by the liver, spleen, and monocyte macrophage system [123]. This suggests that they accumulate in areas of inflammatory response, which can be effectively identified. In the future, this technique could see increased application in diagnosing altitude-related illnesses.
- The references are careless presented.
Response: Thank you for your critical comments. Following the format of references, we have correctly presented the references (pages 23-29, lines 770-1013).
[8] Serra M F, Cotias A C, Pimentel A S, et al. Gold Nanoparticles Inhibit Steroid-Insensitive Asthma in Mice Pre-serving Histone Deacetylase 2 and NRF2 Pathways[J]. Antioxidants, 2022,11(9):1659-1659.
El Alam S, Pena E, Aguilera D, et al. Inflammation in Pulmonary Hypertension and Edema Induced by Hypobaric Hypoxia Exposure[J]. International Journal of Molecular Sciences, 2022, 23(20): 12656-12668.
[28] Yim Y S, Choi J-S, Kim G T, et al. A facile approach for the delivery of inorganic nanoparticles into the brain by passing through the blood–brain barrier (BBB)[J]. Chem. Commun., 2012, 48(1): 61-63.
[29] Li Y, Yang Y, Qing Y A, et al. Enhancing ZnO-NP Antibacterial and Osteogenesis Properties in Orthopedic Ap-plications: A Review[J]. International Journal of Nanomedicine, 2020, 15: 6247-6262.
[69] Ponnanikajamideen M, Rajeshkumar S, Vanaja M, et al. In Vivo Type 2 Diabetes and Wound-Healing Effects of Antioxidant Gold Nanoparticles Synthesized Using the Insulin Plant Chamaecostus cuspidatus in Albino Rats[J]. Canadian Journal of Diabetes, 2019, 43(2): 82-89.
[91] Zeng H, Qi Y, Zhang Z, et al. Nanomaterials toward the treatment of Alzheimer's disease: Recent advances and future trends[J]. Chinese Chemical Letters,2021,32(06):1857-1868.
[92] Konicek D, Leifer F, Chen K-J, et al. Inhaled Treprostinil-Prodrug Lipid Nanoparticle Formulations Provide Long-Acting Pulmonary Vasodilation[J]. Drug Research, 2018, 68(11): 605-614.
[97] Tomomitsu K, Shigeru M, Takuji K, et al. Innovative therapeutic strategy using prostaglandin I2 agonist (ONO1301) combined with nano drug delivery system for pulmonary arterial hypertension[J]. Scientific Re-ports,2021,11(1):7292-7292.
[98] Hade M D, Suire C N, Suo Z. Mesenchymal Stem Cell-Derived Exosomes: Applications in Regenerative Medi-cine[J]. Cells, 2021,10(8):1959-1959.
[118] Haddad F, Mohammed N, Gopalan R C, et al. Development and Optimisation of Inhalable EGCG Nano-Liposomes as a Potential Treatment for Pulmonary Arterial Hypertension by Implementation of the Design of Experiments Approach[J]. Pharmaceutics, 2023,15(2):539-539.

Round 2
Reviewer 3 Report
Comments and Suggestions for Authors
All requested modifications were solved. Paper can be accepted for publication.